

# The role and mechanism of aerobic glycolysis in nasopharyngeal carcinoma

Zhiyong Pan[1,*], Yuyi Liu[1,*], Hui Li[2], Huisi Qiu[1], Pingmei Zhang[1], Zhiying Li[1], Xinyu Wang[1], Yuxiao Tian[1], Zhengfu Feng[1], Song Zhu[1] and Xin Wang[1]

[1] Department of Radiotherapy, Affiliated Qingyuan Hospital, Guangzhou Medical University, Qingyuan, Guangdong, China
[2] Department of Ophthalmology, Affiliated Qingyuan Hospital, Guangzhou Medical University, Qingyuan, Guangdong, China
* These authors contributed equally to this work.

## ABSTRACT

This review delves into the pivotal role and intricate mechanisms of aerobic glycolysis in nasopharyngeal carcinoma (NPC). NPC, a malignancy originating from the nasopharyngeal epithelium, displays distinct geographical and clinical features. The article emphasizes the significance of aerobic glycolysis, a pivotal metabolic alteration in cancer cells, in NPC progression. Key enzymes such as hexokinase 2, lactate dehydrogenase A, phosphofructokinase 1, and pyruvate kinase M2 are discussed for their regulatory functions in NPC glycolysis through signaling pathways like PI3K/Akt and mTOR. Further, the article explores how oncogenic signaling pathways and transcription factors like c-Myc and HIF-1α modulate aerobic glycolysis, thereby affecting NPC's proliferation, invasion, metastasis, angiogenesis, and immune evasion. By elucidating these mechanisms, the review aims to advance research and clinical practice in NPC, informing the development of targeted therapeutic strategies that enhance treatment precision and reduce side effects. Overall, this review offers a broad understanding of the multifaceted role of aerobic glycolysis in NPC and its potential impact on therapeutic outcomes.

## INTRODUCTION

Nasopharyngeal carcinoma (NPC) is a malignancy that originates in the epithelial cells of the nasopharynx, with the epicenter of the tumor frequently situated in the Rosenmüller fossa, from which the neoplasm invades into adjacent anatomical spaces or organs (*Chen et al., 2019*). Distinct from other malignancies, nasopharyngeal cancer exhibits a unique geographical distribution. In 2022, there were 120,416 new cases of nasopharyngeal carcinoma worldwide, accounting for 0.6% of all cancer cases, ranking 23rd among all cancers and the number of deaths was 73,476, accounting for 0.8% of all cancer deaths, ranking 21st (*Bray et al., 2024*). Additionally, the incidence and mortality rates of nasopharyngeal carcinoma in males were higher than in females, with the incidence rate in males being 4.75 times that of females and the mortality rate being 2.5 times that of females

Corresponding authors
Song Zhu, 870182869@qq.com
Xin Wang, xinwang@gzhmu.edu.cn

(*Bray et al., 2024*). The incidence and mortality of NPC have seen a decline in some endemic areas (*e.g.*, Hong Kong, Singapore, and Taiwan) over the past three decades (*Carioli et al., 2017*). However, in other endemic areas, such as certain southern provinces in mainland China, the incidence of NPC has remained stable over the past two decades (*Wei et al., 2010*). According to the World Health Organization, nasopharyngeal carcinoma is categorized into three pathological subtypes: keratinized squamous, non-keratinized, and basal squamous (*Wang et al., 2016*). Non-keratinized nasopharyngeal carcinoma can be bifurcated into differentiated and undifferentiated neoplasms. The keratinized subtype accounts for less than 20% of global cases and is relatively rare in endemic areas such as Southern China. The non-keratinized subtype constitutes the majority of cases (>95%) in endemic areas, and it is primarily associated with Epstein-Barr Virus (EBV) infection (*Wang et al., 2016*; *Pathmanathan et al., 1995*; *Young & Dawson, 2014*). Among these, undifferentiated non-keratinized squamous cell carcinoma is the most prevalent type (*Wang et al., 2016*). This unique geographical distribution is etiologically attributed to genetic susceptibility, environmental factors, and ethnic differences (*Wang et al., 2016*; *Lee et al., 2019*). However, due to its insidious clinical presentation, early-stage nasopharyngeal carcinoma often remains asymptomatic, and most cases are diagnosed as stage II, III, and IV by the time they are detected (AJCC 8th ed.) (*Chen et al., 2021*). The cornerstone of treatment for nasopharyngeal cancer is radiotherapy or combined radiochemotherapy, which boasts a cure rate exceeding 90% in early-stage patients (*Chan, 2011*). However, the prognosis for patients with advanced localized disease and distant metastases remains unsatisfactory. These patients are at a high risk of developing metastasis or local recurrence post-radiotherapy or combined radiotherapy, and they experience significant side effects. Moreover, recent data have indicated that early-stage nasopharyngeal cancer is optimally treated with radiation alone; local and distant recurrences impact the 5-year overall survival rate, which decreases from approximately 90% to 75% in stage I and II patients (*Chan, 2011*). The prognosis is further compromised if nasopharyngeal cancer is diagnosed with cervical or retropharyngeal lymph node metastases (*Mai et al., 2013*). Additionally, patients with nasopharyngeal cancer undergoing radiotherapy have a diminished quality of life, accompanied by severe side effects such as bone marrow suppression (*Caponigro et al., 2010*). However, targeted therapy can accurately identify and treat nasopharyngeal cancer cells with minimal toxicity and side effects, offering a broad clinical application prospect (*Colli et al., 2017*). Increasing evidence suggests that aerobic glycolysis is a critical metabolic characteristic of nasopharyngeal carcinoma. Therefore, targeting key enzymes or regulatory pathways of aerobic glycolysis may represent a promising strategy for the treatment of nasopharyngeal carcinoma.

The hallmarks of cancer encompass six biological capabilities acquired during the multistep development of human tumors. These include sustaining proliferative signaling, evading growth suppressors, resisting cell death, enabling replicative immortality, inducing angiogenesis, and activating invasion and metastasis. These constitute an organizing principle that provides a logical framework for the complexity of neoplastic diseases (*Hanahan & Weinberg, 2011*). Conceptual advances over the past two decades have

introduced two additional emerging hallmarks of potential universality—the reprogramming of energy metabolism and the evasion of immune destruction (*Hanahan & Weinberg, 2011*). Additionally, with the deepening of research, new viewpoints suggest that there may be other potential characteristics and properties. For example, phenotypic plasticity and disrupted differentiation may be a distinct hallmark capability, nonmutational epigenetic reprogramming and polymorphic microbiomes may be unique enabling characteristics that facilitate the acquisition of hallmark capabilities, and senescent cells from different origins may be added to the list of functionally important cell types in the tumor microenvironment (*Hanahan, 2022*). It is well established that under aerobic conditions, normal cells first obtain energy through glycolysis in the cell membrane, followed by mitochondrial oxidative phosphorylation (OXPHOS) to secure a substantial amount of energy. When oxygen is scarce, the cell relies on glycolysis rather than oxygen-consuming mitochondrial metabolism for its energy supply (*Shang, Qu & Wang, 2016*). However, the metabolic pattern of tumors differs from that of normal cells (*Lapa et al., 2020*). In 1920, Otto H. Warburg first discovered that the metabolism of glucose in cancer cells was distinct from that of differentiated cells—even in an aerobic state, tumor cells favored glycolysis over oxidative phosphorylation, which is highly productive, to provide the energy required by tumor cells. He termed this phenomenon "aerobic glycolysis" (later dubbed the "Warburg effect" in 1976 (*Racker, 1976*)), and proposed that "aerobic glycolysis" was the most efficient way to supply energy for tumor cells (*Warburg, Negelein & Posener, 1924*). He attributed this metabolic feature to mitochondrial "respiratory impairment" (*Warburg, 1956*). However, while numerous experimental studies on aerobic glycolysis have been published in the last decades, the detection of intact functional cytochromes in most tumor cells clearly indicates that mitochondrial function is not disrupted (*Vaupel, Schmidberger & Mayer, 2019*). *Warburg (1962)* acknowledged that respiration in cancer cells was not, in fact, impaired, thus undermining his original "impaired respiration" theory (*Vaupel, Schmidberger & Mayer, 2019*). Today's advanced developments in molecular biology and high-throughput molecular analysis have revealed that selective high-rate aerobic glycolysis is due to the accumulation of the products of signaling pathways. These signaling pathways are altered by mutations or changes in gene expression and are often influenced by the tumor microenvironment rather than by mitochondrial dysfunction (*Cairns, Harris & Mak, 2011*). Furthermore, these altered signaling pathways act both independently and in concert with each other during the metabolic reprogramming of the Warburg effect (*Soga, 2013*). Given the infinite proliferation of tumor cells requires a faster energy supply, the rate of ATP production by glycolysis is much faster than oxidative phosphorylation, although ATP production per molecule of glucose by glycolysis is much less efficient (*Domiński et al., 2020*; *Zhang et al., 2020a*). In different tumors, aerobic glycolysis still accounts for 50–70% of the ATP supply (*Mathupala, Ko & Pedersen, 2009*). Emerging evidence suggests that cancer cells are able to suppress anti-tumor immune responses by competing for and depleting essential nutrients or otherwise reducing the metabolic adaptations of tumor-infiltrating immune cells (*Guerra, Bonetti & Brenner, 2020*; *Hurley et al., 2020*). Moreover, the metabolic byproducts generated during aerobic glycolysis serve

as the building blocks for biomolecules, which are indispensable for the accelerated proliferation of tumor cells (*Vaupel, Schmidberger & Mayer, 2019*). Furthermore, lactic acid, a byproduct of aerobic glycolysis, creates an acidic milieu that facilitates cancer invasion and metastasis (*Gatenby & Gawlinski, 2003*). Consequently, aerobic glycolysis is closely related to cancer. The relationship with nasopharyngeal carcinoma has become a research focus in recent years and has attracted extensive academic attention. Although a large number of studies have explored the association between the two, a scoping review to integrate the results in this field is lacking. In this context, we conducted a comprehensive review of the relevance and operation of key enzymes, oncogenic signaling pathways and transcription factors involved in aerobic glycolysis in nasopharyngeal carcinoma.

The review article offers a comprehensive exploration of aerobic glycolysis in nasopharyngeal carcinoma, addressing its role in the tumor's invasiveness and radiosensitivity. The review underscores aerobic glycolysis as a cancer hallmark, particularly amplified in nasopharyngeal carcinoma, affecting processes like invasion and metastasis. It details the regulatory functions of four key enzymes (hexokinase 2, lactate dehydrogenase A, phosphofructokinase 1, and pyruvate kinase M2) that influence tumor progression through pathways like PI3K/Akt and mTOR. This knowledge underpins the development of targeted therapies. The review suggests that targeting components of aerobic glycolysis could enhance treatment precision and reduce side effects. It also proposes using molecular markers related to aerobic glycolysis for prognosis and drug development. Overall, the review is crucial for advancing research and clinical practice in nasopharyngeal carcinoma.

Building on this, the review aims to enlighten scholars with in-depth insights into aerobic glycolysis's mechanisms in nasopharyngeal carcinoma. It provides theoretical support for clinicians, particularly in otolaryngology and oncology, guiding treatment strategies. For academics and educators, it serves as a resource for understanding tumor metabolism and as educational material. Industry researchers will find new directions for drug development through potential therapeutic targets discussed. Funding bodies and policymakers can use this review to inform their decisions, supporting advancements in the field. In essence, the review disseminates extensive knowledge and data, contributing significantly to nasopharyngeal carcinoma research and treatment.

## SURVEY METHODOLOGY

This article presents the results of a comprehensive longitudinal query analysis of four databases: PubMed, Embase, Cochrane Library and Web of Science. The data collection process queried all relevant literature using a combination of keywords: "nasopharyngeal carcinoma and Warburg effect," "nasopharyngeal carcinoma and aerobic glycolysis," "NPC and Warburg effect," and "NPC and aerobic glycolysis." Inclusion and exclusion criteria for the literature were as follows.

### Inclusion criteria

1. Research topic: Studies should focus specifically on the relationship between nasopharyngeal carcinoma (NPC) and aerobic glycolysis, including but not limited to

the specific mechanisms of aerobic glycolysis in NPC, the role of key enzymes, and the regulation of signaling pathways.

2. Research type: Original research papers, review articles, and systematic reviews that provide a detailed analysis or summary of the process of aerobic glycolysis in NPC are included.

3. Experimental design: For experimental studies, *in vitro* or *in vivo* experiments on NPC cells, as well as quantitative analysis of key indicators of aerobic glycolysis (such as glucose uptake, lactate production, ATP generation, *etc.*) are required.

4. Methodology: Scientific and reliable methods should be used to analyze the relationship between NPC and aerobic glycolysis, including molecular biology techniques, metabolomics analysis, immunohistochemistry, *etc.*

5. Data integrity: Studies should provide complete datasets, including detailed experimental procedures, data analysis, and statistical methods, to ensure reproducibility and reliability of the results.

6. Relevance: Articles should be directly relevant to the topic of NPC and aerobic glycolysis, contributing to our understanding of NPC metabolism, therapeutic targets, or clinical significance.

## Exclusion criteria

1. Non-relevance: Studies that are not focused on NPC or aerobic glycolysis will be excluded.

2. Duplicate studies: Topics or results that have been sufficiently explored in previous studies, unless they provide new insights or data, will be excluded.

3. Poor quality: Studies with unscientific designs, unreliable methods, incomplete data, or unsupported conclusions will be excluded.

4 Non-original research: Non-original research (such as case reports, conference abstracts, *etc.*) will not be included unless they provide important supplementary information on the topic of NPC and aerobic glycolysis.

5. Language barriers: Articles published in languages other than English will be excluded to ensure the broad and comparable nature of the review.

6. Unavailability: Articles that cannot be accessed or have incomplete data will be excluded.

In the process of literature screening and evaluation, we adopt a participatory approach involving two or more researchers to ensure the objectivity and accuracy of the screening results. We endeavor to remain impartial and avoid the influence of personal bias on the final conclusions. In case of inconsistency in judgment of certain literature, we will conduct thorough discussion and consultation, and if necessary, invite a third party to arbitrate to ensure the reliability and credibility of the assessment results.

## ENHANCED AEROBIC GLYCOLYSIS IN NASOPHARYNGEAL CARCINOMA

The phenomenon of increased aerobic glycolysis was initially discovered by Warburg and his colleagues over a century ago in rat hepatocellular carcinoma (*Warburg & Minami, 1923*). This heightened aerobic glycolysis has since been observed in a multitude of other cancer types, including nasopharyngeal carcinoma (*Lo et al., 2015*), renal cell carcinoma (*Shuch, Linehan & Srinivasan, 2013*), lung carcinoma (*Feinberg et al., 2017*), breast cancer (*Wu et al., 2020*), gastric carcinoma (*Yuan, 2016*), pancreatic carcinoma (*Xiang et al., 2018*), and prostate carcinoma (*Ciccarese et al., 2016*). According to recent research, cancer cells undergo aerobic glycolysis characterized by *Cairns, Harris & Mak (2011)*, *Soga (2013)*, *Vander Heiden, Cantley & Thompson (2009)*, *Ward & Thompson (2012)*, *Phan, Yeung & Lee (2014)*, *Kato et al. (2018)* (a) an upregulation of glucose transporters and pivotal glycolytic enzymes, an escalation in glycolytic flux, and an accumulation and translocation of glycolytic intermediates, all of which lead to amplified biosynthesis in cancer, (b) a high rate of ATP production to cater to energy requirements, and (c) an accumulation of lactate that propels tumor progression and significantly contributes to tumor acidosis. This, in turn, synergistically bolsters tumor progression, resistance to certain antitumor therapies, and undermines antitumor immunity. Over the past decade, it has been discerned that aerobic glycolysis in nasopharyngeal carcinoma (NPC) is primarily exhibited through glucose uptake, ATP production, and lactate production and accumulation. This is closely associated with the upregulation of the expression of key enzymes implicated in aerobic glycolysis in nasopharyngeal carcinoma cells.

Key glycolytic enzymes are overexpressed or overactive in tumor cells, contributing to the accelerated glycolysis rate (*Li, Gu & Zhou, 2015*). These pivotal enzymes in the glycolytic pathway are upregulated in various types of tumors, thereby exacerbating the aberrations of metabolic pathways in tumor cells. Altenberg et al. have provided a comprehensive overview of the key enzymes involved in glycolysis and their expression status in 24 different cancers, demonstrating an overexpression of these key glycolytic enzymes in head and neck cancers (*Altenberg & Greulich, 2004*). There is mounting evidence to suggest that these key enzymes involved in aerobic glycolysis could potentially serve as viable therapeutic strategies against nasopharyngeal cancer. There are four key enzymes implicated in the glycolytic pathway: Hexokinase (HK), lactate dehydrogenase A (LDHA), phosphofructokinase 1 (PFK1), pyruvate kinase type M2 (PKM2) (*Li, Gu & Zhou, 2015*) (Fig. 1).

### Hexokinase (HK) enhances aerobic glycolysis in nasopharyngeal carcinoma

HK is the first rate-limiting enzyme in the glycolytic pathway. It catalyzes the phosphorylation of glucose to glucose-6-phosphate using adenosine triphosphate (ATP) (*Tan & Miyamoto, 2015*). In mammals, there are four known isozymes of HK (HK1-4) (*Wilson, 2003*), and a fifth isozyme, HexoKinase Domain Containing Protein 1 (HKDC1),

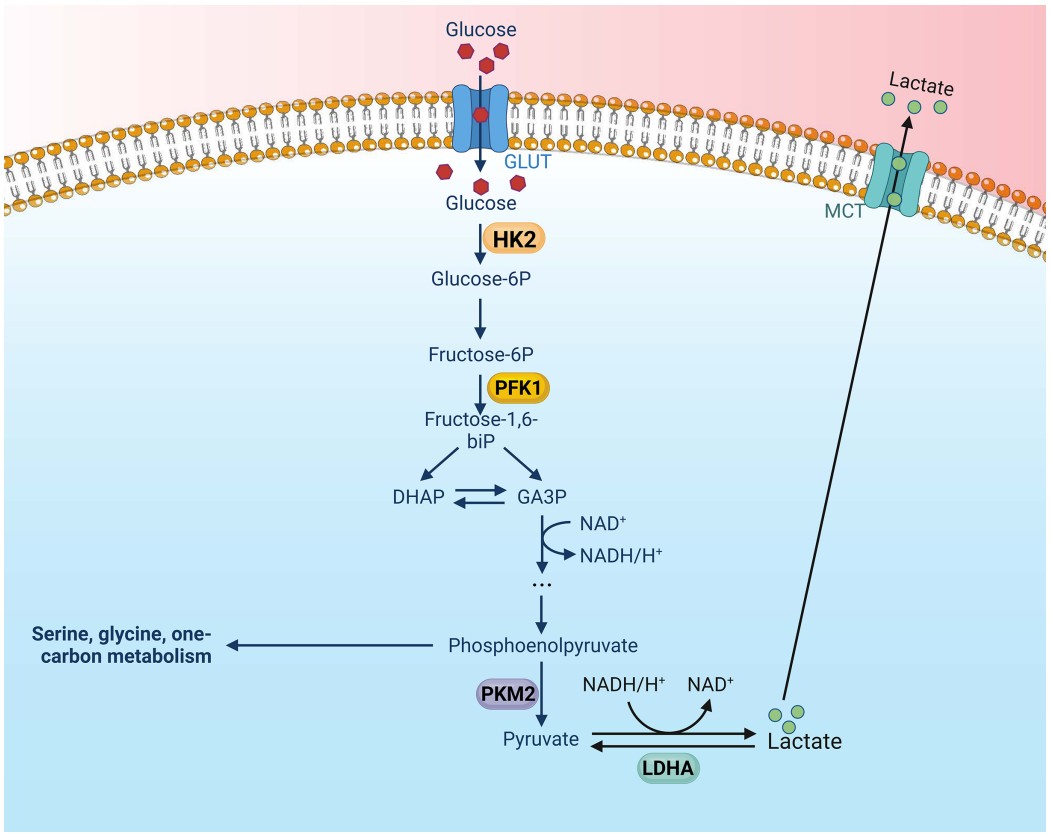

**Figure 1 Aerobic glycolysis, along with its four key enzymes.** Figure created with bioRender.com.

has been recently identified (*Irwin & Tan, 2008*). HK1-3 and HKDC1 are 100 kDa proteins; HK1 is ubiquitously expressed in all types of mammalian cells and is the predominant form of HK in most tissues. Although widely present, HK2 is expressed at lower levels than HK1 and is the primary isoenzyme of insulin-sensitive tissues, including adipose tissue, skeletal muscles and heart. The expression patterns of HK3 and HKDC1 are more restricted and not as well characterized (*Garcia, Guedes & Marques, 2019*). HK4, an enzyme with a catalytic molecule of 50 kDa, is expressed in high amounts in hepatocytes and pancreatic β-cells, where it is involved in glucose metabolism and insulin secretion. Among the isozymes, HK2 is more potent than the others in promoting aerobic glycolysis. The mechanisms by which HK2 enhances glycolysis have been the subject of extensive research (*Gong et al., 2012*). First, HK2 interacts with and binds to voltage-dependent anion-selective channel protein 1 in the mitochondrial outer membrane. The interaction enhances ATP production and inhibits apoptosis by activating ATP-synthesis-related enzymes (*Xu & Herschman, 2019*). Secondly, when HK2 binds to voltage-dependent anion-selective channel protein 1, it is shielded from inhibition by downstream products like G-6-P. This enhances glycolysis and speeds up ATP production (*Mathupala, Ko & Pedersen, 2009*).
Existing research indicates that the amplified expression of hexokinase fosters aerobic glycolysis in cells of nasopharyngeal carcinoma. A study conducted by *Xu et al. (2021)* revealed that miR-9-1 directly curtailed HK2 expression by latching onto the target site of HK2's 3′-β UTR, thereby impeding glycolytic metabolism in NPC cells. They discovered a significant correlation between the expression level of miR-9-1 and factors such as patient gender, T-stage, TNM stage, locoregional recurrence-free survival (LRRFS), and overall survival (OS). In 2020, *Li et al. (2020)* reported that INSL5 upregulated HK2 expression in nasopharyngeal carcinoma, thereby enhancing aerobic glycolysis. Overexpression of HK2 has been linked with adverse outcomes such as stage progression, poor prognosis, metastasis, and/or treatment resistance in various malignant tumors, including hepatocellular carcinoma, lung cancer, breast cancer, colorectal carcinoma, prostate cancer, glioblastoma, and diffuse large B-cell lymphoma (*Ciscato et al., 2021*). Moreover, a comprehensive gene expression examination revealed that the transcript levels of hexokinase in nasopharyngeal carcinoma could be augmented by the induction of LMP1 (*Lincet & Icard, 2015*). LMP1 is widely recognized as a viral oncogene in nasopharyngeal carcinoma (*Tsao et al., 2002*), and it has been found to induce elevated HK expression in NPC cells to stimulate aerobic glycolysis (*Xu et al., 2021*; *Cai et al., 2017*; *Xiao et al., 2014*). In 2014, *Xiao et al. (2014)* reported that the weakening of the PI3-K/Akt-GSK3beta-FBW7 signaling axis mediated by LMP1 led to an upregulation of HK2, which subsequently enhanced aerobic glycolysis in NPC. Furthermore, this study discovered a positive correlation between HK2 and LMP1 in NPC biopsy specimens. High levels of HK2 were significantly associated with poor overall survival of NPC patients following radiation therapy, and were indicative of a poor prognosis for NPC patients post-radiation therapy (*Xiao et al., 2014*). Therefore, the elevated expression of HK2 is strongly linked with NPC patients. Targeting HK2 could potentially be a novel strategy to enhance the effectiveness of radiotherapy for NPC. It also presents a promising metabolic target for the development of innovative therapies for NPC.

## 6-Phosphate fructose-1-kinase (PFK-1), PFKFB2 and PFKFB3 kinases enhance aerobic glycolysis in nasopharyngeal carcinoma

PFK-1, the second rate-limiting enzyme of glycolysis, facilitates the conversion of fructose 6-phosphate to fructose 1,6-diphosphate and adenosine diphosphate (ADP). This four-subunit mutase is inhibited by ATP concentration and phosphoenolpyruvate (PEP) mutation, but is activated by fructose-2,6-bisphosphate (F26BP) (*Okar et al., 2001*). In humans, there are three PFK genes: PFKL in the liver, PFKM in muscle, and PFKP in platelets. According to the specific metabolic requirements of tissues, the proportions of these isoforms may vary (*Hasawi, Alkandari & Luqmani, 2014*). The fructose-2 and 6-bisphosphatase/6-phosphofructo-2-kinase (PFKFB) enzyme is a bifunctional enzyme encoded by four different genes (PFKFB 1-4) that stimulates the expression of the F26BP protein (*Kim et al., 2006a*). However, it has been observed that increased PFK-1 activity promotes glycolysis and proliferation in cancer cells (*Yalcin et al., 2009*). Interestingly, as a key regulator of tumor growth, metastasis, and glycolysis, PFKFB3 is closely regulated by HIF-1α, AKT, and PTEN (*Panigrahy et al., 2002*). Moreover, PKFB3 is frequently

overexpressed in numerous human tumors, including those of the lung, breast, colon, pancreas, ovary, and thyroid (*Atsumi et al., 2002*).

PFK1, PFKFB2, and PFKFB3 are found to be overexpressed in nasopharyngeal carcinoma cells. A study conducted by *Li et al. (2020)* demonstrated that the activation of STAT5 by INSL5, and its subsequent binding to the promoter of PFK1, enhanced PFK1 transcription. This, in turn, promoted the expression of glycolytic genes in nasopharyngeal carcinoma cells. In 2017, *Su et al. (2017)* discovered that JMJD2A upregulated the expression of the PFK gene in NPC, thereby enhancing aerobic glycolysis. Moreover, PFKFB2 and PFKFB3 have been shown to be closely related to glycolysis. In a 2021 study by *Qi et al. (2021)*, it was found that the expression of CENP-N was positively correlated with the expression of PFKFB2 and PFKFB3 in NPC. Furthermore, they found that the knockdown of CENP-N suppressed the expression of genes related to glucose metabolism, thereby inhibiting glucose metabolism (*Qi et al., 2021*). In a 2017 study, *Cai et al. (2017)* discovered that the expression levels of PFKFB2 and PFKFB3 were down-regulated in NPC cells following the knockdown of CENP-N. This led to a decrease in the level of aerobic glycolysis. *Sung et al. (2017)* found in their 2017 study that LAMP1 induced an increase in the transcriptional level of PFKFB2. Similarly, *Yalcin et al. (2014)* found that LAMP1 induced an increase in the transcriptional level of PFKFB2 and PFKFB3 in their 2017 study. Both studies demonstrated the potential for upregulation of PFKFB2 and 3 to promote the Warburg effect. Moreover, PFKFB3 has the ability to translocate to the nucleus of nasopharyngeal carcinoma cells, thereby modulating cell cycle protein-dependent kinase (CDK) activity (*Yalcin et al., 2014*). This results in an arrest in the cell cycle and an inhibition of cellular death. Therefore, enhanced expression of the PFK gene promotes nasopharyngeal carcinoma progression, and targeting PFKFB2, PFKFB3 and PFK1 could be a potential therapeutic approach for treating NPC.

## PK enhances glycolysis in nasopharyngeal carcinoma

PK catalyzes the rate-limiting enzyme in the final step of glycolysis (*Harris, Mccracken & Mak, 2012*). It facilitates the transfer of the phosphate group of phosphoenolpyruvate (PEP) to ADP, yielding a pyruvate molecule and an ATP molecule (*van Niekerk & Engelbrecht, 2018*). There are four isoforms of PK, which include the erythrocytic (PKR) and hepatic (PKL) forms, as well as the muscle isoforms M1 (PKM1) and M2 (PKM2) (*Méndez-Lucas et al., 2017*). Liver, kidneys, intestines, and pancreas express PKL predominantly, while erythrocytes express PKR exclusively. The PKM gene has two exons that can be spliced selectively into two products, PKM1 and PKM2 (*Clower et al., 2010*). PKM1 isoforms are found in tissues with high catabolic demands, such as muscle, heart, and brain. In contrast to the constitutively expressed PKM1, the isoform PKM2 is tightly regulated. It is predominantly expressed in the brain and liver, is highly upregulated in cancer cells, and correlates with a poor prognosis (*Shang, Qu & Wang, 2016*). PKM2 activity is regulated by its oligomerization status, multiple metastable effectors, and post-translational modifications (*Clower et al., 2010*). PKM2 exists in two forms. One is a tetramer, which is located in the cytoplasm and has a high catalytic activity. This form allows for the rapid conversion of PEP to pyruvate, facilitating glycolytic flux and more

ATP (*Anastasiou et al., 2012*). A monomer or dimer with low catalytic activity is the other form. There is evidence that this form can act as a co-activator of many transcription factors, including NF-κB, HIF-1α, β-linker protein/c-Myc, and STAT3 (*van Niekerk & Engelbrecht, 2018*; *Azoitei et al., 2016*). Upon entering the nucleus, PKM2 induces transcription of its target gene. For example, it promotes vascular endothelial growth factor A (VEGF-A), PKM2, LDH-A, and HIF-1α-targeted expression of GLUT. This promotes angiogenesis, positive feedback-regulated glycolysis, and cancer cell growth (*Luo et al., 2011*).

Recent studies have revealed an enhanced expression of PKM2 in nasopharyngeal carcinoma cells. PKM2 is a downstream target of HIF-1 and interacts with HIF-1 to reprogram glucose metabolism by transactivating the expression of other glycolytic genes, such as glucose transporter1. This enhances glucose uptake and increases lactate dehydrogenase (LDH), leading to an increase in the conversion of pyruvate into lactate (*Luo et al., 2011*; *Fukuda et al., 2007*; *Gao et al., 2012*; *Gogvadze, Zhivotovsky & Orrenius, 2010*; *Kim et al., 2006b*; *Papandreou et al., 2006*; *Wheaton & Chandel, 2011*). As a result, aerobic glycolysis in nasopharyngeal carcinoma is enhanced. In a 2017 study, *Sung et al. (2017)* found that PKM2 expression was enhanced by LMP1 in nasopharyngeal carcinoma cells by activating HIF-1α, which promoted aerobic glycolysis in these cells. Additionally, it was found that decreased PKM2 expression in NPC cells inhibited NPC cell proliferation and invasion. In a 2021 study, *Qi et al. (2021)* discovered that the knockdown of CENP-N led to the downregulation of genes such as PKM2 in NPC cells. This affected aerobic glycolysis, cell cycle, apoptosis and cell proliferation in NPC cells. *Zhang et al. (2020b)* found that TET2 inhibited PKM dimer formation in nasopharyngeal carcinoma cells and glycolysis in NPC cells by interacting with PKM. The NPC cells were subsequently inhibited in their proliferation and invasion as a result. In summary, the enhanced expression of PKM2 is closely related to the proliferation and invasion of NPC cells. Therefore, targeting PKM2 presents an extremely attractive potential target for NPC therapy.

## Lactate dehydrogenase-A (LHD-A) enhances aerobic glycolysis in nasopharyngeal carcinoma

LDH catalyzes the interconversion of pyruvate and lactate. This reaction is accompanied by the interconversion of nicotinamide adenine dinucleotide (NADH) and NAD+ when oxygen is absent or in short supply (*Fisher, 2001*). LDH is a tetramer composed of two different subunits, LDHA and LDHB, which can be assembled into five different combinations (LDH1-LDH5) (*Markert, Shaklee & Whitt, 1975*). LDHB is ubiquitously expressed in humans and is the major isoform found in cardiac muscle. LDHA, on the other hand, is the major isoform found in skeletal muscle and other highly glycolytic tissues. LDHA is encoded by the LDHA gene and is usually present as a tetramer (LDH-S). Its primary function is to convert pyruvate to lactate and NADH to NAD+ (*Dawson, Goodfriend & Kaplan, 1964*). While LDHA is primarily located in the cytoplasm, it has also been found in mitochondria and the nucleus (*Reddy & Shukla, 2000*; *Fiume et al., 2014*; *Brooks et al., 1999*). Outside the nucleus, LDHA plays a key role in glycolysis. However,

within the nucleus, LDHA functions as a single-stranded DNA-binding protein (SSB) and may be involved in DNA replication and transcription (*Feng et al., 2018*). Notably, LDHA has a higher affinity for pyruvate and a higher Vmax (maximum reaction rate at a given amount of enzyme) for pyruvate reduction compared to LDHB. Aberrant expression and activation of LDHA has been found to be strongly associated with several cancers (*Feng et al., 2018*; *Le et al., 2010*; *Fantin, St-Pierre & Leder, 2006*). Therefore, LDHA is considered a promising target for cancer prevention and treatment.

In nasopharyngeal carcinoma, high expression of LDHA suggests enhanced proliferation, migration, and invasion. Jumonji C domain 2 (JMJD2A) belongs to the group of histone demethylases, which are overexpressed in many types of cancer (*Whetstine et al., 2006*). In a 2017 study, *Su et al. (2017)* found that nasopharyngeal carcinoma cells activated LDHA expression through transcription of JMJD2A. This promoted aerobic glycolysis and the proliferation, migration, and invasive capacity of NPC cells (*Su et al., 2017*). Furthermore, elevated LDHA levels in most tumors imply high glycolysis and suggest high proliferation, high migration, high invasive capacity and poor prognosis of these human malignancies (*Koukourakis et al., 2005*, *2003*, *2009*). Notably, LDHA affects the activity of key enzymes of aerobic glycolysis through increased expression in nasopharyngeal carcinoma, which in turn enhances aerobic glycolysis. For example, *Lo et al. (2015)* found in a 2015 study that LMP1 promotes aerobic glycolysis by increasing the uptake of glucose and glutamine in nasopharyngeal carcinoma cells. This enhances LDHA activity and lactate production, which in turn decreases pyruvate kinase (PK) activity and pyruvate concentration (*Lo et al., 2015*). This study also found that LDHA was enhanced by increased expression levels of the PI3K/AKT upstream pathway FGFR1 and its ligand FGF2, which directly enhanced aerobic glycolysis expression. Moreover, inhibiting LDHA expression in nasopharyngeal carcinoma can impede the progression and metastasis of NPC. For instance, a 2016 study by *Li et al. (2016)* found that LDHA was a direct target of the miR-34 b/c cluster and miR-449a. Overexpression of the miR-34b/c cluster and miR-449a inhibited LDHA transcription, which in turn inhibited NPC cell proliferation, invasion, and migration (*Li et al., 2016*). Research studies in other cancer types have also found evidence that targeting LDHA is an attractive cancer treatment strategy. For example, selective knockdown of the LDHA gene has been found to inhibit anchored growth in several transformed cell lines (*Shim et al., 1997*) and *in vivo* growth of transplanted breast tumors (*Fantin, St-Pierre & Leder, 2006*). In summary, the expression of LDHA is closely related to the biological behavior of nasopharyngeal carcinoma, making LDHA a very promising target for nasopharyngeal carcinoma.

## ROLE OF AEROBIC GLYCOLYSIS IN NASOPHARYNGEAL CARCINOMA

### Aerobic glycolysis promotes proliferation, growth and immune escape of nasopharyngeal carcinoma cells

Aerobic glycolysis, a novel hallmark of NPC, is believed to promote NPC induction of immune evasion, growth, and proliferation through several mechanisms: (a)

Overexpression of key glycolytic enzymes and glucose transporters, accelerated glycolytic fluxes, and the subsequent accumulation of glycolytic intermediates that are diverted to biomass synthesis in nasopharyngeal carcinoma cells (*Lo et al., 2015*). The intermediates can be incorporated into different metabolic pathways to synthesize nucleotides, lipids, and proteins, which promote cancer cell proliferation (*Vander Heiden, Cantley & Thompson, 2009*). Cancer cells also meet the precursor requirements for nucleic acid biosynthesis by increasing aerobic glycolysis to promote glutamine catabolism (*Ganapathy-Kanniappan, 2018*). (b) ATP is generated at a high rate through aerobic glycolysis to meet the increased energy requirements of cancer cells, allowing tumors to adapt to their energy-deficient microenvironment (*Xu & Herschman, 2019*); (c) accumulation of high levels of lactate drives tumor progression. Lactate production and hydrogen ion generation, which are generated during these processes, result in acidification of the extracellular environment, which suppresses immunity. It has been reported that lactate contributes to immune escape from different pathways of immune escape, and its suppression of immune cell function further promotes cancer cell survival and contributes largely to tumor acidosis. This in turn synergistically promotes tumor progression and resistance to impairs antitumor immunity and certain antitumor therapies (*Koukourakis et al., 2005*). Increased LDHA expression in nasopharyngeal carcinoma not only increases lactate production and glucose utilization, but also increases ATP utilization, which decreases ATP levels in nasopharyngeal carcinoma cells, resulting in inhibition of nasopharyngeal carcinoma cell growth and proliferation (*Su et al., 2017*).

In the aforementioned studies of aerobic glycolysis in nasopharyngeal carcinoma, enhancement or inhibition of the expression of key enzymes of aerobic glycolysis (*e.g.*, PKM2, LDHA) has been found to be closely associated with nasopharyngeal carcinoma growth (*Su et al., 2017*; *Qi et al., 2021*; *Sung et al., 2017*; *Zhang et al., 2020b*; *Li et al., 2016*). In addition, *Luo et al. (2018)* found that down-regulation of PTE gene expression by LMP1 mediated by DNMT1 could enhance the activity of AKT rate-limiting enzyme and inhibit oxidative phosphorylation of the aerobic glycolysis pathway in nasopharyngeal cancer cells, and these alterations have important roles in the growth of nasopharyngeal cancer cells (*Luo et al., 2018*).

## Aerobic glycolysis promotes nasopharyngeal carcinoma cell invasion and metastasis, angiogenesis

Owing to the inherent characteristics of tumorous tissues, which are predisposed to hypoxic conditions, cancerous cells are inclined to metastasize to alternate locations in order to augment their energy and blood supply, thereby ensuring their survival. Aerobic glycolysis facilitates this metastasis and invasion in NPC predominantly through the acidification of the extracellular environment mediated by lactate and H+ (*Hanahan & Weinberg, 2011*; *Dhup et al., 2012*; *Hirschhaeuser, Sattler & Mueller-Klieser, 2011*; *Justus, Dong & Yang, 2013*), This involves several key processes: (1) In the extracellular environment, the low pH instigates the disruption of normal tissues through cysteine asparaginase-mediated or p53-dependent apoptosis (*Williams, Collard & Paraskeva, 1999*). (2) Acidification of the extracellular matrix (ECM) stimulates the secretion of

protein hydrolases such as histone B or metalloproteinases, which assist in ECM degradation and expedite metastasis (*Gatenby & Gillies, 2004*). (3) Low pH levels also cause immunosuppression, which allows metastatic cancer cells to evade immune system surveillance, leading to persistent metastasis (*Lardner, 2001*). (4) In cancer, acidification of the extracellular environment by lactate and H+ promotes the secretion of vascular endothelial growth factor (VEGF) and interleukin 8, both of which are angiogenic factors that induce angiogenesis (*Shi et al., 2001*; *Jung et al., 2011*). Consequently, the extracellular structural alterations and immunosuppression induced by aerobic glycolysis render cancer cells prone to metastasis and invasion. Nasopharyngeal carcinoma cells induce angiogenesis by enhancing HIF-1/VEGF activity, which subsequently promotes the activation of glycolytic enzymes (*Denko, 2008*). Consequently, this amplified aerobic glycolysis plays a pivotal role in modulating extracellular structures and immunosuppression in nasopharyngeal carcinoma, thereby rendering cancer cells susceptible to metastasis, invasion, and angiogenesis.

## Aerobic glycolysis promotes tumor microenvironment in nasopharyngeal carcinoma

The microenvironment of nasopharyngeal carcinoma (NPC) is intricate and exhibits significant heterogeneity, with its constituents falling into cellular and non-cellular categories. The cellular components encompass (*Arneth, 2019*; *Liu et al., 2019b*; *Tao et al., 2017*; *Byun et al., 2017*) (a) Tumor endothelial cells, which are formed to transport nutrients, oxygen, and blood, and to establish a network of lymphatic vessels; (b) cancer-associated fibroblasts, which contribute to tumor growth, invasion, survival and carcassing, and (c) immune cells, which participate in immune responses within the tumor microenvironment (TME). Conversely, non-cellular components include the extracellular matrix, which fosters tumor development, metastasis, and progression (*Spill et al., 2016*; *Eble & Niland, 2019*). The acidic environment is a quintessential feature of the tumor microenvironment and plays a pivotal role in carcinogenesis during tumor development (*Massa et al., 2017*). Lactate, a byproduct of aerobic glycolysis, significantly contributes to acidosis in the cellular microenvironment due to the dynamic shuttling of lactate and protons from cancer cells to extracellular sites. It's noteworthy that the nasopharynx serves as one of the primary organs of defense against viral and bacterial entry and infection, rendering its underlying microenvironment highly heterogeneous and immunogenic prior to malignant transformation (*Binnewies et al., 2018*). There is a substantial immune infiltrate in the NPC tumor microenvironment, including monocytes, B cells, dendritic cells, and T cells. It is largely necessary for lymphoid NPC progression to produce inflammatory cytokines. In contrast, chronic inflammatory response in NPC can induce the expression of several downstream targets such as up-regulation of HIF-1α expression. This results in the production of large amounts of lactate and H+ due to the enhancement of aerobic glycolysis in NPC (*Dawson, Port & Young, 2012*; *Lai et al., 2010*; *Middeldorp & Pegtel, 2008*; *Morris et al., 2008*), thereby promoting the formation of an extracellular acidic environment.

# REGULATORY MECHANISMS OF AEROBIC GLYCOLYSIS IN NASOPHARYNGEAL CANCER

The hallmark difference between cancer cells and normal cells is metabolic reprogramming, and aerobic glycolysis is considered to be the major metabolic phenotype in cancer (*Hanahan & Weinberg, 2011*). The aberrant activation of signaling pathways can lead to a myriad of human diseases. Concurrently, a series of aberrant transmembrane signaling pathways, including pro-survival (PI3K/Akt, MAPK, Wnt/β-catenin, NF-κB, STAT3) and pro-apoptotic (p53, endoplasmic reticulum stress) pathways, are currently under investigation in nasopharyngeal cancer. These have been associated with the prognosis, onset, and progression of nasopharyngeal carcinoma by affecting biological processes such as apoptosis, the cell cycle, and DNA repair. However, in cancer cells, PI3K/AKT, AMPK, mTOR, Wnt and MAPK signaling pathways are implicated in regulating aerobic glycolysis (*Cai et al., 2018*; *Engelman, Luo & Cantley, 2006*; *Han et al., 2015*; *Yoshida, 2015*; *Irey et al., 2019*) (Fig. 2). In the following sections, we will delve into how aerobic glycolysis is regulated in NPC through PI3K/AKT, mTOR signaling pathways and related transcription factors (c-Myc and HIF-1).

## PI3K/AKT pathway regulates aerobic glycolysis in nasopharyngeal carcinoma

The PI3K/AKT signaling pathway is recognized for its significant role in inducing glucose metabolism in cancer cells. Phosphatidylinositol-3-kinases (PI3K) are a family of signaling enzymes that include three major classes of lipid kinases, I–III (with class I further subdivided into Ia and Ib) and the distantly related class IV (*Engelman, 2009*). PIP3, which is crucial for Akt activation, is formed by phosphorylating phosphatidylinositol 4,5-bisphosphate (PIP2) and converting it to phosphatidylinositol 3,4,5-triphosphate (PIP3) (*Engelman, Luo & Cantley, 2006*). Akt, also called protein kinase B, is an essential serine/threonine protein kinase that is directly activated by PI3K. PI3K-dependent AKT directly phosphorylates and activates PFK2, enhancing the production of fructose-2,6-bisphosphate. This ultimately activates the rate-limiting enzyme of glycolysis PFK1 (*Novellasdemunt et al., 2013*; *Lee et al., 2018*). AKT is required for a variety of cellular processes including cell growth, metabolism, and survival (*Nicholson & Anderson, 2002*). In numerous studies, this pathway has been demonstrated to be altered in many cancers and regulates a number of critical cellular functions, such as proliferation, invasion, apoptosis, metastasis, glucose homeostasis and angiogenesis (*Engelman, Luo & Cantley, 2006*; *Guerrero-Zotano, Mayer & Arteaga, 2016*).

The PI3K/Akt pathway can modulate the activity or expression of some key enzymes of glycolysis in NPC to influence glycolysis, such as HK2, LDHA, and PDHK1. For instance, LMP1 further stabilizes the oncoprotein c-Myc, a key regulator of LMP1-mediated HK2 up-regulation in NPC, by decreasing the activity of glycogen synthase 3b (GSK3b) through activation of the PI3K/AKT pathway (*Koppenol, Bounds & Dang, 2011*). In other words, LMP1 can promote the glycolysis process by activating the PI3K/AKT pathway and ultimately enhancing HK2 signaling (*Xiao et al., 2014*). Additionally, LMP1 increases the

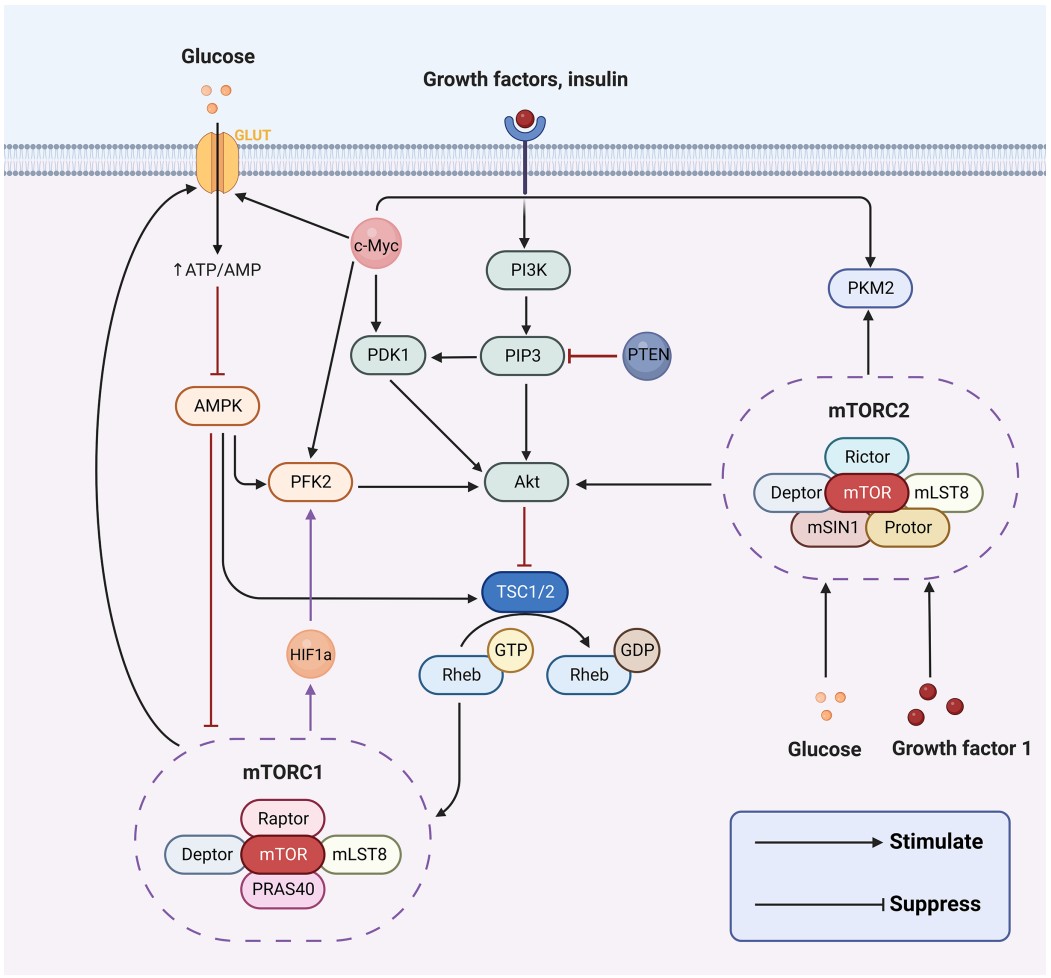

**Figure 2 Aerobic glycolysis can be regulated by a variety of transcription factors and many signaling pathways.** Figure created with bioRender.com.

expression of fibroblast growth factor 1 and its ligand fibroblast growth factor 2 (FGF2) and activates one of the upstream pathways of PI3K/AKT, thereby promoting glycolysis. This potentiates the activities of hypoxia-inducible factor 1α (HIF-1a), LDHA, and pyruvate dehydrogenase kinase 1 (PDHK1), but inhibits PKM2 isoform and pyruvate dehydrogenase A1 (PDHA1) activity (*Warburg & Minami, 1923*). Similarly, Latent membrane protein 2 (LMP2), encoded by EBV, is a transmembrane protein with two isoforms (LMP2A and LMP2B). LMP2A recruits Lyn/Syk kinases through its ITAM motifs (Y74/Y85) and Y112 residues, activating the PI3K/Akt pathway, which collectively drives the expression of key glycolytic enzymes such as HK2 and LDHA (*Dawson, Port & Young, 2012*). Moreover, 3-phosphatidylinositol-dependent protein kinase 1 (PDK1) is the first node of PI3K signaling output, and it phosphorylates AKT at the Thr308 site, which is a prerequisite for full AKT activation (*Yang et al., 2022b*). *Cai et al. (2018)* found that when Chibby (a β-conjugated protein antagonist) was overexpressed in nasopharyngeal carcinoma cells, PDK1 protein expression was significantly decreased. This led to a decrease in the phosphorylation of AKT, which in turn regulated a decrease in aerobic

glycolysis in nasopharyngeal carcinoma cells (*Cai et al., 2018*). *Yang et al. (2022a)* discovered that FOXM1 directly binds to the PDK1 promoter region and elevates PDK1 expression at the transcriptional level. This results in the enhancement of the PI3K/AKT pathway, which subsequently promotes lactate production, ATP generation, and aerobic glycolysis (*Cai et al., 2018*). Furthermore, elevated PDK1 expression has been observed in various human cancers, such as colorectal cancer (*Xu et al., 2015*), breast cancer (*Maurer et al., 2009*), hepatocellular carcinoma (*Wu et al., 2016*), and esophageal cancer (*Yang et al., 2014*). Overexpression of PDK1 leading to increased activation of the PI3K/AKT pathway has been associated with a poor prognosis in these tumors. Activation of the PI3K/Akt pathway plays a pivotal role in a variety of cellular functions, including cell differentiation, proliferation, and intracellular trafficking (*López-Knowles et al., 2010*). These functions are involved in cancer development. Thus, the PI3K/AKT signaling pathway influences aerobic glycolysis in nasopharyngeal carcinoma cells by directly or indirectly regulating the expression of certain enzymes in the glycolytic pathway.

## The mTOR pathway regulates aerobic glycolysis in nasopharyngeal carcinoma

The mammalian target of rapamycin (mTOR) pathway, a key regulator of cellular growth and *in vivo* homeostasis, adeptly orchestrates anabolic and catabolic processes, growth factors, energy levels, oxygen supply, and nutrient availability (*Leung et al., 2014*). However, neoplastic cells can overexpress specific glycolytic enzymes that manipulate the mTOR signaling pathway to preserve clonality and foster migration (*Huang et al., 2018*; *Vijayakrishnapillai et al., 2018*). In recent times, a wealth of evidence has accentuated the cardinal role of mTOR as a central hub within malignant cells, coordinating a plethora of essential enzymes to support neoplastic proliferation and extracellular nutritional balance (*Mossmann, Park & Hall, 2018*). The activation of mTOR has been shown to invigorate glucose metabolism, mRNA transcription, protein biosynthesis, and lipid synthesis, all of which are vital to cellular proliferation (*Dowling et al., 2010*). mTOR, a fundamental serine/threonine protein kinase within the PI3K family, comprises two unique catalytic subunit protein complexes: mTOR complex 1 (mTORC1) and mTOR complex 2 (mTORC2) (*Hara et al., 2002*; *Kim et al., 2002*; *Saxton & Sabatini, 2017*). These complexes have been recognized as playing instrumental roles in modulating metabolism within cancerous cells (*Saxton & Sabatini, 2017*). mTORC1 is often hyperactivated in glycolytic cancer cells, suggesting that targeting factors upstream of mTORC1 may be more effective in obstructing glycolysis within cancer cells than inhibiting factors downstream thereof (*Saxton & Sabatini, 2017*). mTORC2 can be stimulated by growth factors such as insulin (*Zoncu, Efeyan & Sabatini, 2010*). Recent explorations have unveiled that mTORC2 can also be stimulated by glucose, glutamine, and mTORC1 (*Byun et al., 2017*; *Javary et al., 2017*). The mTORC2 pathway is indispensable for maintaining normal glucose homeostasis and metabolic regulation *in vivo*, and hyperactivation of mTORC2 may be implicated in the onset of tumorigenesis (*Yuan et al., 2018*).

It has been demonstrated that the mTOR signaling pathway is intricately linked to nasopharyngeal carcinogenesis. In nasopharyngeal carcinoma cells, the mTOR signaling

pathway fosters cell proliferation and tumor invasion by inducing sterol regulatory element binding protein 1 (SREBP1)-mediated lipid synthesis (*Lo et al., 2018*). The activation of insulin-like growth factor 1-mTORC2 has been shown to propel nasopharyngeal carcinoma metastasis with reprogrammed glucose metabolism (*Zhang et al., 2019*). The Akt/mTOR axis may be inhibited by sodium butyrate (NaBu) by promoting EGFR degradation in a manner mediated by histone deacetylase 6, which decreases the likelihood of metastasis in nasopharyngeal carcinoma cells (*Huang et al., 2019*). Furthermore, LMP1 promotes aerobic glycolysis in nasopharyngeal carcinoma cells by activating the mTORC1 pathway and the NF-kB pathway, resulting in elevated transcriptional activity of glucose transporter-1 (*Zhang et al., 2019*). LMP1 has been further discovered to augment glucose metabolism by activating a glucose metabolizing enzyme, PDHE1a, *via* the mTORC2 signaling pathway (*Zhang et al., 2019*). The mTOR signaling pathway has been shown to be intimately associated with glycolytic enzymes. One of the upstream regulators of mTOR is the PI3K/Akt signaling pathway. The inhibition of PI3K/mTOR has been demonstrated to effectively obstruct the membrane localization of GLUT1, which is crucial for glycolytic expression levels (*Godoy et al., 2006*). Consequently, the mTOR signaling pathway is typically upregulated in NPC and can foster the proliferation and metastasis of nasopharyngeal carcinoma cells by enhancing the expression of glucose transporters and key enzymes of glycolysis. Therefore, targeting the mTOR pathway emerges as an efficacious strategy for the treatment of NPC.

## c-Myc regulates aerobic glycolysis in nasopharyngeal carcinoma

c-Myc is an oncogenic protein that is swiftly degraded by the ubiquitin-proteasome system in non-transformed cells, and it plays a role in ribosome biogenesis for cell metabolism, growth, proliferation, apoptosis, and tumorigenesis (*van Riggelen, Yetil & Felsher, 2010*). The oncogene c-Myc is known to positively regulate aerobic glycolysis through several mechanisms. First of all, in nasopharyngeal carcinoma cells, c-Myc serves as a pivotal transcription factor that regulates aerobic glycolysis by modifying the activities of multiple key enzymes of aerobic glycolysis. It can transactivate and enhance the expression of LDH (*Grüning, Lehrach & Ralser, 2010*), and also directly boost the expression of PKM2, HK2, and GLUT1, thereby promoting glycolytic flux (*Dang et al., 2008*; *Kim et al., 2007*). For instance, LMP1 promotes aerobic glycolysis and epithelial cell transformation by activating the FGF2/FGFR1 signaling pathway and augmenting c-Myc expression, thus suggesting that transcription factors play a role in the pathogenesis of EBV-driven NPC (*Lo et al., 2015*). Secondly, when c-Myc binds to the PKM2 promoter, it not only induces PKM2 expression but also promotes the PKM2/PKM1 ratio, thereby fostering cancer cell survival. Thirdly, nuclear PKM2 can also function as a coactivator for β-catenin, which in turn induces c-Myc expression, forming a positive feedback loop that promotes the continued expression and maintenance of glycolytic genes (*Yang et al., 2012*). Fourthly, c-Myc can synergize with HIF-1α to activate PDK1, leading to increased the production of lactate and the extracellular environment's acidity (*Kim et al., 2007*). c-Myc also interacts in a complex manner with other transcription factors and signaling pathways such as β-catenin, HIF-1α, ERK signaling pathway signaling pathway and JAK/STAT3. This may

collectively promote aerobic glycolysis in nasopharyngeal carcinoma. Moreover, c-Myc is also intimately associated with cancer cell transformation. According to studies, c-Myc mediates glutamine catabolism by promoting glutamine uptake and glutamine catabolism, thereby preserving the mitochondrial tricarboxylic acid (TCA) cycle's integrity to promote cancer cell survival (*Dang, Le & Gao, 2009*), and c-Myc function can be enhanced by mutation of the gene itself or by upstream oncogenic pathways induced c-Myc expression (*Grüning, Lehrach & Ralser, 2010*). Given the critical role of c-Myc in NPC carcinogenesis, it is evident that c-Myc emerges as an attractive target for the development of novel therapies.

## HIF-1$\alpha$ regulates aerobic glycolysis in nasopharyngeal carcinoma

HIF-1 consists of two subunits, the constitutively expressed HIF-1β and the rate-limiting HIF-1α. Two significant structural domains, the oxygen-dependent degradation and transcriptional activation structural domains, are located within the HIF-1α protein molecule (*Wang et al., 1995*). HIF-1α is a pivotal regulator of glycolytic metabolism, and in hypoxic environments, HIF-1α expression is induced. It can be stabilized by hypoxia and then bind to hypoxia-responsive elements in target promoters, resulting in transcription of relevant genes involved in the process of overcoming hypoxia (*Okar et al., 2001*). Hypoxia is a prevalent characteristic of many solid tumors due to the rapid proliferation and expansion of cancer cells, and activation of the HIF-1 transcription factor is the most recognized mechanism of pathway acquisition of hypoxic cells in these tumors (*Masoud & Li, 2015*). Activated HIF-1 regulates the transcription of numerous target genes, which subsequently participate in crucial biological processes, including cell proliferation, metastasis, angiogenesis, glucose metabolism, and resistance to chemotherapy and radiation. In recent years, experimental evidence has demonstrated that in the development of nasopharyngeal carcinoma, HIF-1α promotes tumor cell proliferation and growth, provides cells with a rapid source of energy and biosynthetic precursors (*Sung et al., 2017*), and its up-regulation has been associated with poor prognosis of nasopharyngeal carcinoma (*Lo et al., 2015*). By inducing enzymes of the glycolytic pathway, including HKII, PFK1, LDHA, aldolase, and GLUT-1 and GLUT-3, HIF-1 shifts glucose metabolism in hypoxic tumor cells to the glycolytic pathway, and this metabolic shift leads to a shift in energy production (*Denko, 2008*). Additionally, transactivation of genes such as PDK1 and MAX-interacting protein 1 (MXI1) by HIF enables down-regulation of mitochondrial function and enhancement of aerobic glycolysis (*Denko, 2008*).

HIF-1α is commonly upregulated in NPC patients. As a pivotal regulator of glucose metabolism in NPC, its activation plays an important role in the regulation of the Warburg effect, primarily at the transcriptional level, as follows. First of all, HIF-1α promotes the transcription of GLUT-1, which enhances glucose uptake by glycolysis (*Yao et al., 2018*). Secondly, HIF-1α can up-regulate the transcription of several glycolytic enzymes, including PFK1, PKM2, HK2, LDHA, and glycerol triphosphate dehydrogenase, which further enhances the level of glycolysis (*Marin-Hernandez et al., 2009*). For instance, *Sellam et al. (2020)* found that in patients with nasopharyngeal carcinoma, HIF-1α

regulated lactate dehydrogenase expression and enhanced glycolytic flux through LDH-A overexpression. Thirdly, HIF-1α promotes PDK expression and inhibits PDH activity, leading to the conversion of pyruvate to lactate and facilitating the glycolytic pathway (*Anastasiou et al., 2012*). For example, *Yang et al. (2022a)* found that in nasopharyngeal carcinoma development, HIF-1α was able to activate glycolysis in nasopharyngeal carcinoma cells by inducing the expression of the key enzyme of glycolysis, PDK1. Fourthly, tumor microenvironmental factors are able to induce stabilization of HIF-1α and increase the expression of FOXM1, a transcription factor that promotes glycolysis through transcriptional activation of PDK1 promoter activity (*Yang et al., 2022a*). *Sung et al. (2017)* found in their 2017 study that HIF-1α was able to regulate the expression of multiple target genes, including PDK1 and PKM2, to promote the glycolytic pathway in nasopharyngeal carcinoma. As well, in addition to hypoxic stress, HIF-1α could also be affected by signaling pathways, such as PI3K/Akt/mTOR, AMPK, and Raf/MAPK, leading to accelerated levels of glycolysis (*Deberardinis et al., 2008*). *Lo et al. (2015)* found that LMP1, through activation of the FGF2/FGFR1 signaling pathway and increasing the expression of c-Myc and HIF-1α, promotes aerobic glycolysis and epithelial cell transformation. Direct regulation of HIF-1α by LMP1 has also been reported. LMP1 significantly increased HIF-1α protein and mRNA expression by decreasing tristetraprolin (TTP) and pumilio RNA-binding family member 2 expression levels. LMP1's carboxy-terminal activating regions 1 (CTAR1) and 3 (CTAR3) are involved in this program. The CTAR1 is involved with the extracellular signal-regulated kinase 1 and 2 pathways (ERK1/2) and is also part of the ERK1/2/NF-kB (nuclear factor kB) pathway that enhances HIF-1α activity. The CTAR3 is involved with STAT3 (*Sung et al., 2016*). Notably, HIF-1 is an important regulator of glucose metabolism and increased activation of JNKs/c-Jun signaling by promoting LMP1 to enhance HIF-1/VEGF activity and induce angiogenesis (*Denko, 2008*). Therefore, HIF-1α may emerge as a very promising metabolic target for NPC therapy.

## CONCLUSIONS

Cancer research has progressed in many ways. There is growing evidence that cell plasticity can influence tumor development, growth, metastasis, dormancy, and treatment (*Pérez-González, Bévant & Blanpain, 2023*). In cancer therapy, cell plasticity can lead to drug resistance, and various strategies can target cell plasticity to eliminate drug-resistant cells and treat tumors (*Pérez-González, Bévant & Blanpain, 2023*). Moreover, the tumor microbiome plays a crucial role in cancer immunotherapy by influencing cancer progression and immune responses through various mechanisms, including direct interaction with immune cells, production of metabolites, and interaction with immunotherapies such as immune checkpoint inhibitors (*Yang et al., 2024*). Cellular senescence can, in certain circumstances, promote tumor development and malignant progression. Senescent cells can influence various capabilities of cancer cells by releasing the senescence-associated secretory phenotype (SASP) (*Birch & Gil, 2020*). Tumor-associated fibroblasts and endothelial cells may also undergo senescence, thereby affecting the tumor microenvironment (*Birch & Gil, 2020*). Additionally, the role of

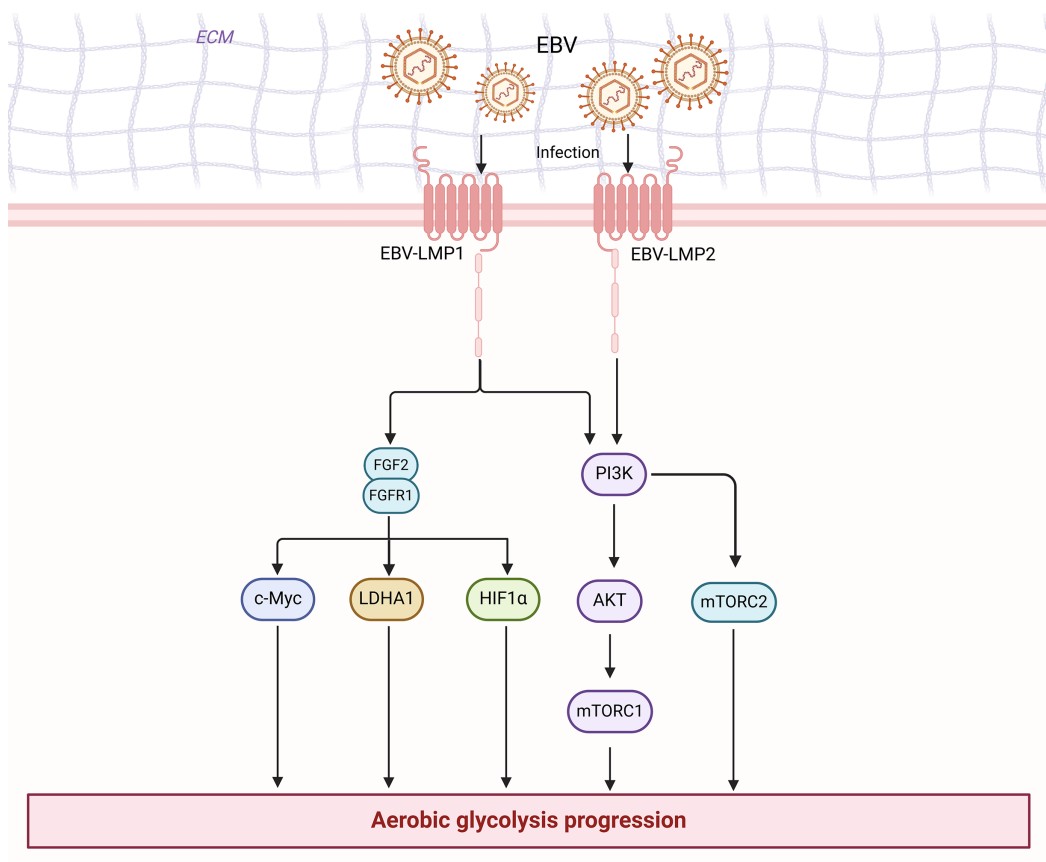

**Figure 3 EBV is associated with enhanced aerobic glycolysis in nasopharyngeal carcinoma cells.** Figure created with bioRender.com.

non-mutational epigenetic reprogramming in cancer development has garnered attention. The abnormal properties of the tumor microenvironment can alter the epigenome to promote cancer cell growth (*Hanahan, 2022*). The Warburg effect, a key metabolic characteristic of cancer induced by active metabolic reprogramming (*Koppenol, Bounds & Dang, 2011*), presents numerous potential therapeutic targets for NPC. However, no article has systematically addressed the role and mechanism of aerobic glycolysis in nasopharyngeal carcinoma. We have reviewed their relationship and mechanisms for the first time. In summary, we found a close relationship between EBV and enhanced aerobic glycolysis in nasopharyngeal carcinoma cells (Fig. 3). With an understanding of the significant role of aerobic glycolysis in NPC, further comprehension of the regulatory mechanisms involved will provide opportunities to develop new therapeutic approaches. These potential therapeutic strategies include targeting key enzymes in glycolysis (HK, PFK, PK, and LDHA) to inhibit glucose uptake or lactate levels, which in turn directly inhibits the aerobic glycolytic process, and targeting glycolytic signaling pathways (PI3K/ AKT and mTOR) and transcription factors (HIF-1 and c-myc) to indirectly attenuate glycolytic expression. However, there are still significant challenges in the current treatment of tumors against aerobic glycolysis, including how to precisely target key signaling pathways and enzymes in tumor cells and how to avoid adverse effects on normal

**Table 1 Targeted inhibitor of key enzymes of aerobic glycolysis.**

| Metabolic pathway | Targetbased actions | Drug name | Ref |
|---|---|---|---|
| Key enzyme in aerobic glycolysis | HK2 inhibitor | 3-Bromopyruvate | Zou et al. (2015) |
| | | 2-DG | Lee, Ko & Yun (2018) |
| | PDK1 inhibitor | DCA | Zhang, Zhou & Xiao (2020) |
| | PKM2 inhibitor | Shikonin | Zhang, Zhou & Xiao (2020) |
| | LDHA inhibitor | Oxamate | Zhai et al. (2013) |

cells. Studies on the molecular mechanisms of aerobic glycolysis in nasopharyngeal carcinoma are still limited.

Currently, there are widely used drugs that target key enzymes of aerobic glycolysis (*e.g.*, HK 2, PKM 2, PDHK1, and LDHA) to inhibit glycolysis (Table 1). For instance, 2-deoxy-D-glucose (2-DG), a glucose analog, can be converted by HK 2 into 2-deoxy-D-glucose-6-phosphate (2-DG-6P), which inhibits HK 2 noncompetitively (*Lee, Ko & Yun, 2018*). It has been reported that 2-DG inhibits aerobic glycolysis in NPC cells and suppresses the growth, metastasis, and invasion of nasopharyngeal carcinoma (NPC) cells when used alone (*Li et al., 2020*; *Cai et al., 2017*), but there is still a lack of clinical trials combining 2-DG with chemotherapeutic agents for the treatment of NPC. 3-Bromopyruvic acid (3-BP), a different type of HK 2 inhibitor, is able to inhibit the HK 2 activity directly, thereby strongly inhibiting the glycolytic process. Metabolic inhibitors such as 3-bromopyruvate and oxamate (a competitive LDH-A inhibitor) have been reported to inhibit HK and LHDA activities in tumors and activate macrophages and T cells in preclinical models of NPC (*Zou et al., 2015*; *Zhai et al., 2013*). Dichloroacetic acid (DCA) exhibits promising therapeutic effects on metabolic targets by inhibiting PDHK1 activity in aerobic glycolysis and bolstering T cells, thereby suppressing tumor metabolism. Shikonin can be utilized to inhibit the activity of PKM2, resulting in a significant reduction in glucose uptake, lactate release, and ATP levels by shikonin in a dose-dependent manner in nasopharyngeal carcinoma cells (*Zhang, Zhou & Xiao, 2020*). Shikonin can greatly diminish glucose uptake, lactate release, and ATP levels in nasopharyngeal carcinoma cells in a dose-dependent manner.

In addition, there have been studies on aerobic glycolytic pathways and related transcription factors. For instance, cetuximab-targeted drugs inhibit aerobic glycolysis in nasopharyngeal carcinoma by targeting EGFR and reducing the expression of the PI3K/Akt pathway. The dual PI3K-mTOR inhibitor, PF-0469150210, was found to significantly reduce the weight of nasopharyngeal carcinoma tumors in mice through the inhibition of the PI3K/AKT/mTOR pathway, but had little effect on the body weight of mice, and no serious toxicity was observed during treatment (*Wong et al., 2013*). Brevilin A inhibits the PI3K/Akt/mTOR and STAT3 signaling pathways *in vitro*, and Brevilin A treatment resulted in no significant weight loss in treated mice (*Liu et al., 2019a*). This suggests that PF-04691502 and Brevilin A could be used as preclinical development of chemotherapeutic agents for nasopharyngeal carcinoma. Evofosfamide is a

hypoxia-activated prodrug that selectively targets hypoxic regions of nasopharyngeal carcinoma tissues, and Evofosfamide, as a single agent in combination with DDP, targets the selectively hypoxic portion of nasopharyngeal carcinoma, thereby reducing the overexpression of HIF-1α in nasopharyngeal carcinoma tissues to inhibit aerobic glycolysis (*Huang et al., 2018*). Nonetheless, there is a dearth of clinical trials concerning the inhibition of aerobic glycolytic pathways in nasopharyngeal carcinoma. Moreover, burgeoning evidence suggests that the utilization of anti-PD-1 and anti-PD-L1 monoclonal antibodies as a therapeutic strategy for NPC patients is efficacious (*Chen et al., 2013*; *Hsu et al., 2017*). Targeting the EBV antigens EBNA 1, LMP 1, and LMP 2 can be accomplished through the infusion of EBV-specific cytotoxic T cells (EBV-CTL), and clinical investigations have demonstrated the safety and effectiveness of this approach (*Louis et al., 2010*). Intriguingly, silencing the Fas gene in EBV-CTL cells may aid in the clearance of cytotoxic T cells by NPC cells (*Dotti et al., 2005*). Consequently, amalgamating metabolic drugs with immunotherapeutic agents may assist in surmounting immunotherapy resistance, and the impact of conjoining aerobic glycolysis and immunity in the treatment of nasopharyngeal carcinoma patients could potentially become a viable clinical alternative.

Prospective directions may include the development of more precise and effective drugs, as well as the enhancement of therapeutic efficacy through combination therapy. Furthermore, a profound comprehension of the regulatory mechanisms of aerobic glycolysis in neoplastic cells will also contribute to the creation of innovative therapeutic strategies. In summation, aerobic glycolysis is instrumental in NPC proliferation, immune evasion, invasion and metastasis, angiogenesis, among others. Targeting the pivotal factors involved in aerobic glycolysis (*e.g.*, inhibition of key enzymes and regulation of pathways) represents a potential novel therapeutic paradigm for the treatment of NPC.

## ACKNOWLEDGEMENTS

The authors want to thank bioRender.com for providing the graphic platform.

### Funding
This research was funded by the National Natural Science Foundation of China OF Song Zhu, grant number 82002920. The funders had no role in study design, data collection and analysis, decision to publish, or preparation of the manuscript.

### Grant Disclosures
The following grant information was disclosed by the authors:
National Natural Science Foundation of China: 82002920.

### Competing Interests
The authors declare that they have no competing interests.

## Author Contributions

- Zhiyong Pan conceived and designed the experiments, performed the experiments, analyzed the data, prepared figures and/or tables, and approved the final draft.
- Yuyi Liu conceived and designed the experiments, performed the experiments, analyzed the data, prepared figures and/or tables, and approved the final draft.
- Hui Li performed the experiments, prepared figures and/or tables, and approved the final draft.
- Huisi Qiu performed the experiments, prepared figures and/or tables, and approved the final draft.
- Pingmei Zhang performed the experiments, prepared figures and/or tables, and approved the final draft.
- Zhiying Li performed the experiments, prepared figures and/or tables, and approved the final draft.
- Xinyu Wang performed the experiments, prepared figures and/or tables, and approved the final draft.
- Yuxiao Tian performed the experiments, prepared figures and/or tables, and approved the final draft.
- Zhengfu Feng performed the experiments, prepared figures and/or tables, and approved the final draft.
- Song Zhu conceived and designed the experiments, analyzed the data, prepared figures and/or tables, authored or reviewed drafts of the article, and approved the final draft.
- Xin Wang conceived and designed the experiments, analyzed the data, prepared figures and/or tables, authored or reviewed drafts of the article, and approved the final draft.

## Data Availability

This is a literature review.

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
