# Peer review of "The role and mechanism of aerobic glycolysis in nasopharyngeal carcinoma"

_PeerJ, doi:10.7717/peerj.19213_

## Round 0.1 · original submission · Minor Revisions

The reviewers found your manuscript well-written, thorough and comprehensive. However, they identified some issues that need to be addressed to improve your manuscript. The reviewers suggested including a section on the limitations of the current research and on future directions as well as a figure explaining the relationship between aerobic glycolysis in NPC and Epstein-Barr Virus. Lastly, references should be added in table 1.

Please, submit a detailed rebuttal which shows where and how you have taken all comments and suggestions into consideration. If you do not agree with some of the reviewers’ comments or suggestions, please explain why. Your rebuttal will be critical in making a final decision on your manuscript. Please, note also that your revised version may enter a new round of review by the same or by different reviewers. Therefore, I cannot guarantee that your manuscript will eventually be accepted.

·

Basic reporting

The manuscript offers a well-structured review of aerobic glycolysis in nasopharyngeal carcinoma (NPC). It begins with a concise abstract and a thorough introduction, providing background on NPC and the Warburg effect. The survey methodology is clearly outlined, including inclusion and exclusion criteria used in the study.

The main body of the review is logically organized. It contains sections on pyruvate kinase (PK) and lactate dehydrogenase-A (LDHA), insights into their roles in NPC metabolism. The article effectively progresses from basic mechanisms to clinical implications, covering key enzymes, regulatory pathways, and potential therapeutic targets. The review article offers a different view compared to recent review articles with its specific focus on key enzymes and therapeutic targets.

Experimental design

Pan et al. have conducted a comprehensive literature search across multiple databases, which strengthens the review's breadth and depth. The study design is appropriate for a review article, with a thorough synthesis of current knowledge. The inclusion of both basic science and clinical research enhances the review's scope and relevance.

Validity of the findings

The findings presented appear to be valid and well-supported by the cited literature. The authors have critically analyzed and synthesized information from multiple studies, providing a balanced view of the current state of knowledge in the field. They also highlight areas where further research is needed.

General Comments:
Strengths:
- Comprehensive coverage of the topic
- Clear structure and logical flow of information
- Detailed discussion of key enzymes and their roles in NPC
- Clear linkage between basic science and potential clinical applications

Areas for improvement:
1. Please consider revising the sentences in lines 40 and 210 to enhance readability, "Cochrane library" should be capitalized in line 137, some acronyms (e.g., LRRFS and OS) are used without first being spelled out in full. Keep sentences short where possible
2. The sentence "There are four isoforms of PK, which include erythrocytic (PKR) the and hepatic (PKL) forms..." contains a grammatical error. Please revise.
3. The sentence "In nasopharyngeal carcinoma cells by enhancing HIF-1/VEGF activity..." is incomplete and needs restructuring.
4. Given the rapid pace of cancer research, consider including a section on recent advances or future directions supported with literature from the past 3-5 years.
5. Add the relevant references to the table 1

Reviewer 2 ·

Basic reporting

1. The article used excellent English writing
2. The background used sufficient references to support the importance of this study
3. The article structure, figures, and tables were excellent
4. The article was under the journal's scope
5. This article has not been reviewed recently
6. The introduction clearly explained the subject and could motivate the audience

Experimental design

1. This article is a review article, the contents were within the journal's aims and scope
2. The method section clearly explains the investigation and methods of this study, and there is no ethical concern.
3. Every statement is supported by the reference and the statements were already well paraphrased. However, the reference should use the latest version (for instance, GLOBOCAN 2022 has been published)
4. The review was organized well and every section was interconnecting

Validity of the findings

There were excellent findings in this study and the findings could answer the problem stated in the introduction. Finally, the conclusion could resolve the gap. However, to make it more useful, it is better if the authors add an additional figure to explain the relationship between oxidative glycolysis with NPC and EBV.

Additional comments

None

---

## Round 0.2 · Minor Revisions

Your revised manuscript was reviewed by both original reviewers. Both reviewers agreed that you thoroughly addressed their comments, which greatly improved your manuscript. One of the reviewers suggested that you include information/evidence on the role of LMP2, since it was mentioned in the conclusions and included in figure 3.

Please, submit a detailed rebuttal which shows where and how you have taken all comments and suggestions into consideration. If you do not agree with some of the reviewers’ comments or suggestions, please explain why. Your rebuttal will be critical in making a final decision on your manuscript. Please, note also that your revised version may enter a new round of review by the same or by different reviewers.

Reviewer 1 ·

Basic reporting

The authors have addressed the questions raised following the first review. The manuscript is acceptable and will add value to the scientific community.

Experimental design

The authors have addressed the questions raised following the first review. The manuscript is acceptable and will add value to the scientific community.

Validity of the findings

The authors have addressed the questions raised following the first review. The manuscript is acceptable and will add value to the scientific community.

Additional comments

None

Reviewer 2 ·

Basic reporting

These sections were adjusted, and better than previous version

Experimental design

The methods and scope were clear

Validity of the findings

This section was good. However, in the conclusion section, there was a statement about "LMP 2". In the previous section, there is no explanation about this term.

Additional comments

The authors should add an explanation about LMP 2 and its relation to glycolysis. Therefore, the figure 3 will be supported by certain evidence.

---

## Round 0.3 · accepted · Accept

Thank you for addressing the reviewer comments and thus greatly improving your manuscript.

Reviewer 2 ·

Basic reporting

Clear and unambiguos

Experimental design

The methods and scope were clear

Validity of the findings

Excellent

Additional comments

None